# Towards Stochastic Gradient Variance Reduction by Solving a Filtering Problem

**Xingyi Yang**
National University of Singapore
xyang@u.nus.edu

## Abstract

Stochastic gradient descent is commonly used to optimize deep neural networks, but it often produces noisy and unreliable gradient estimates that hinder convergence. To address this issue, we introduce **Filter Gradient Descent** (FGD), a family of stochastic optimization algorithms that consistently estimate the local gradient by solving an adaptive filtering problem. By incorporating historical states, FGD reduces the variance in stochastic gradient descent and improves the current estimation. We demonstrate the efficacy of FGD in numerical optimization and neural network training, where it outperforms traditional momentum-based methods in terms of robustness and performance. Code is available at https://github.com/Adamdad/Filter-Gradient-Decent.

## 1 Introduction

Stochastic gradient descent (SGD) is one of the most common techniques to solve the optimization problem. Consider training a parametric model $f(\cdot; \boldsymbol{\theta})$ parameterized by $\boldsymbol{\theta} \in \mathbb{R}^d$ over a finite set of training data $\mathcal{D} = \{(\mathbf{x}_i, y_i)\}_{i=1}^N$. At iteration $t$, a subset of the dataset $\mathcal{D}_t \subseteq \mathcal{D}$ is randomly sampled. Denote $\boldsymbol{g}_t = \nabla_{\boldsymbol{\theta}} J(\boldsymbol{\theta}_t; \mathcal{D}_t)$ as the observed gradient, we minimize the objective by updating the weight $\boldsymbol{\theta}$ in the down-hill direction with step size $\eta$

$$\boldsymbol{\theta}_{t+1} = \boldsymbol{\theta}_t - \eta \boldsymbol{g}_t \tag{1}$$

However, SGD suffers from two critical problems: it is highly greedy and noisy. The sampled data points $\mathcal{D}_t$ may not represent the overall data statistics well, leading to unreliable gradient estimates. Additionally, SGD only considers the local error surface, resulting in large variance of the estimated gradient and slower convergence with inferior performance Wang et al. (2013).

In this paper, we take the stochastic gradient as a special signal and solve the filtering problem to establish a *best estimate* for the system from the incomplete and potentially noisy observations:

*Given past gradients $\{\boldsymbol{g}_i\}$ for $0 \leq i \leq t$, what is the best estimate $\hat{\boldsymbol{g}}_{t|t}$ of the true gradient $\hat{\boldsymbol{g}}_t$ of the system based on past observations?*

Inspired by the successful application of filters in signal denoising and reconstruction, we introduce a general framework called **Filter Gradient Decent** (FGD), a remedy to the noisy gradient with the filters to reduce variance and make consistent estimations of the gradient in stochastic optimization. It takes the stochastic gradient as a special signal and solves the filtering problem or denoising problem to estimate the true gradient direction under noisy observations.

## 2 Gradient estimation with filters

In this paper, we make a bold analogy between gradient and high-dimensional signal and reliably estimate gradient by solving a filtering problem. Given the current and past noisy gradient observation $\{\boldsymbol{g}_i\}_{i=0}^t$, we estimate the true state of the current gradient $\hat{\boldsymbol{g}}_{t|t}$ using filter $F$

$$\hat{\boldsymbol{g}}_{t|t} = F(\boldsymbol{g}_t, \boldsymbol{g}_{t-1}, \ldots, \boldsymbol{g}_0) \tag{2}$$

The estimated gradient then is utilized to update the model weight $\boldsymbol{\theta}$ via Equation 1. Algorithm 1 shows the overall framework. We name it **Filter Gradient Decent**.

The filter selection depends on the parametric form between the true state and the observed noisy gradient. We select four well-known filters (1) Moving Average filter (FGD-MA) (2) Auto-regressive filter (FGD-AR) (3) Kalman filter (FGD-K) Kalman (1960); Yang (2021) and (4) Wavelet filter (FGD-W) Strang & Nguyen (1996) as $F$ and would like to study how different filters affect the gradient estimation. The detailed formulations are in the Appendix.

---

**Algorithm 1:** Filter Gradient Decent

Initial Network Parameter with $\boldsymbol{\theta}_0$ and Filter $F$;
**Result:** Optimal Network Parameter $\boldsymbol{\theta}$
**for** $t = 0, \ldots, T-1$ **do**
    Compute Gradient:
        $\boldsymbol{g}_t = \nabla_{\boldsymbol{\theta}} J(\boldsymbol{\theta}_t; \mathcal{D}_t)$;
    Filtering Step:
        $\hat{\boldsymbol{g}}_{t|t} = F(\boldsymbol{g}_t, \boldsymbol{g}_{t-1}, \ldots, \boldsymbol{g}_0)$;
    Gradient Descent:
        $\boldsymbol{\theta}_{t+1} = \boldsymbol{\theta}_t - \eta \hat{\boldsymbol{g}}_{t|t}$;

---

## 3 EXPERIMENT

We demonstrate FGD with various filter designs can efficiently solve numerical optimization and image classification problems. The experiment details are listed in the Appendix for reference.

**Study 1: Non-convex Optimization.** We first solve a non-convex problem using (1) SGD (2) Adam Kingma & Ba (2014) (3) FGD-MA with order 1 and 2 (4) FGD-AR with order 1 (momentum) and 2 (5) FGD-K and (6) FGD-W. In this experiment, we intend to minimize a non-convex function

$$f(x, y) = (x^2 + y^2) + 5\sin(x + y^2) \tag{3}$$

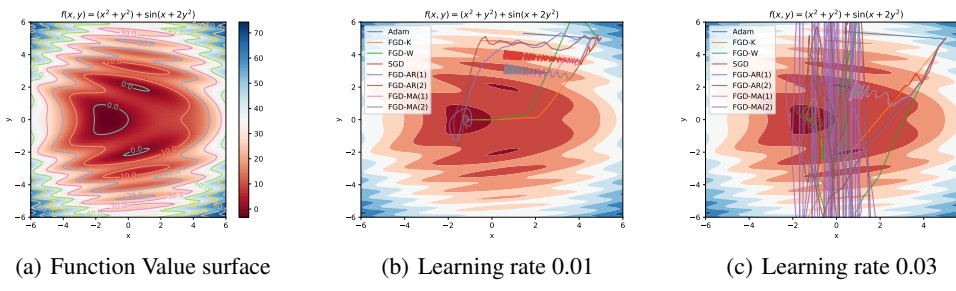

| (a) Function Value surface | (b) Learning rate 0.01 | (c) Learning rate 0.03 |

Figure 1: Experiment for Numerical Non-convex optimization

Figure 1 visualizes the function value surface. We also show the optimization trajectories in Figure 1 Mid and Right. FGD resolves the local minimum problem by boosting the lowpass band of the gradient. Under large and small learning rates, SGD was trapped at a local minimum. On the other hand, FGD is able to find the global solution constantly with smoothed gradient. This shows that, compared with vanilla SGD which only relies on local gradient, FGD can robustly solve the non-convex optimization problem with filtered gradients.

**Study 2: Multi-layer Perceptron.** We train a multi-layer perceptron (MLP) on MNIST LeCun et al. (2010) dataset using (1) SGD (2) Adam Kingma & Ba (2014) (3) FGD-MA with order 1 and 2 (4) FGD-AR with order 1 (momentum) and 2 (5) FGD-K and (6) FGD-W.

Table 1 shows the test accuracy given various parameter setups on MNIST. **First**, FGD outperforms SGD with a small learning rate and large batch size. **Another observation** from the table is that FGD-MA and FGD-K are robust to all kinds of hyper-parameter settings. **In addition**, FGD-MA with order 1 and order 2 makes the best performance on average. Compared with the traditional momentum method, it provides a cheap solution to training the neural network with great potential.

Table 1: Test Accuracy on MNIST with MLP with different hyper-parameter setting

| Setting | | Adam | FGD-K | FGD-W | SGD | FGD-AR(1) | FGD-AR(2) | FGD-MA(1) | FGD-MA(2) |
|---------|---|------|-------|-------|-----|-----------|-----------|-----------|-----------|
| Batch 4 | lr 0.001 | 91.55 | 85.41 | 90.16 | 86.79 | 90.60 | 90.93 | 88.96 | 88.92 |
| Batch 16 | lr 0.001 | 89.95 | 80.59 | 87.58 | 77.90 | 88.85 | 88.87 | 84.38 | 84.38 |
| Batch 64 | lr 0.001 | 88.55 | 36.79 | 81.02 | 35.86 | 85.89 | 85.88 | 50.63 | 50.61 |
| Batch 4 | lr 0.01 | 85.55 | 91.41 | 91.48 | 90.41 | 91.18 | 90.42 | 91.34 | 90.73 |
| Batch 16 | lr 0.01 | 90.71 | 89.09 | 91.46 | 88.71 | 91.66 | 91.61 | 89.68 | 89.51 |
| Batch 64 | lr 0.01 | 87.86 | 86.22 | 89.26 | 85.96 | 90.22 | 90.16 | 89.41 | 89.41 |
| Batch 4 | lr 0.1 | 11.52 | 91.29 | 20.53 | 88.07 | 26.52 | 26.16 | 91.48 | 85.94 |
| Batch 16 | lr 0.1 | 12.03 | 90.64 | 89.54 | 91.77 | 71.81 | 79.36 | 92.13 | 92.07 |
| Batch 64 | lr 0.1 | 30.31 | 81.80 | 90.94 | 90.18 | 89.67 | 89.14 | 91.15 | 91.23 |
| Avg. | | 65.34 | 81.47 | 81.33 | 81.74 | 80.71 | 81.39 | **85.46** | 84.76 |

URM STATEMENT

The authors acknowledge that at least one key author of this work meets the URM criteria of ICLR 2023 Tiny Papers Track. Author Xingyi Yang meets the URM criteria of ICLR 2023 Tiny Papers Track because his age is under 30 and not located in North America, Western Europe and UK, or East Asia.

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

# A APPENDIX

## A.1 RELATED WORK

### A.1.1 FILTERS

In signal processing, a filter is a device or process that removes some unwanted components or features from a signal. We introduce four types of filters used in this paper and their basic concepts.
**Moving-average Filter**. In time series analysis, the moving-average (MA) model specifies that the output variable depends linearly on the current and various past values of a stochastic term. $MA(q)$ refers to the moving average model of order q, where the output $\mathbf{y}_t$ is a weighted linear combination of $\mathbf{x}_t$ and $q$ past input $\{\mathbf{x}_{t-1}, \ldots, \mathbf{x}_{t-p}\}$

$$\mathbf{y}_t = \mathbf{x}_t + \sum_{i=1}^{q} a_i \mathbf{x}_{t-i} \tag{4}$$

**Autoregressive Filter**. An autoregressive (AR) model is a type of random process that the output variable depends linearly on its own previous values and on a stochastic term. $AR(p)$ indicates an autoregressive model of order p, where the output $\mathbf{y}_t$ is a weighted linear combination of input $\mathbf{x}_t$ and $p$ past output value $\{\mathbf{y}_{t-1}, \ldots, \mathbf{y}_{t-p}\}$

$$\mathbf{y}_t = \mathbf{x}_t + \sum_{i=1}^{p} b_i \mathbf{y}_{t-i} \tag{5}$$

**Kalman Filter**. The Kalman filter Kalman (1960) addresses the general problem of trying to estimate the state $\mathbf{x} \in \mathbb{R}^n$ of a discrete-time controlled process that is governed by the linear stochastic difference equation with a measurement $\mathbf{y} \in \mathbb{R}^m$

$$\mathbf{x}_t = \mathbf{F}_{t-1}\mathbf{x}_{t-1} + \mathbf{I}_{t-1}\mathbf{u}_t + \mathbf{w}_{t-1}, \quad \mathbf{y}_t = \mathbf{C}_{t-1}\mathbf{x}_t + \mathbf{v}_t \tag{6}$$

$\mathbf{F}_t \in \mathbb{R}^{n \times n}$ is the state transition matrix which relates the state to the previous step, $\mathbf{I}_t \in \mathbb{R}^{n \times k}$ is the control matrix that relates the optional control input $\mathbf{u} \in \mathbb{R}^k$ to current state, and $\mathbf{C}_t \in \mathbb{R}^{m \times n}$ is measurement matrix. The random variables $\mathbf{w}_t \sim \mathcal{N}(0, Q)$ and $\mathbf{v}_t \sim \mathcal{N}(0, R)$ represent the process and measurement noise respectively. They are assumed to be independent of each other, white, and with normal probability distributions.

The Kalman filter solves the problem in two steps: the prediction step, where the next state of the system is predicted given the previous measurements, and the update step, where the current state of the system is estimated given the measurement at that time step. The steps translate to equations as follows

- Prediction Step

$$\hat{\mathbf{x}}_{t|t-1} = \mathbf{F}_{t-1}\mathbf{x}_{t-1} + \mathbf{I}_t\mathbf{u}_t \tag{7}$$

$$\hat{\mathbf{P}}_{t|t-1} = \mathbf{F}_{t-1}\mathbf{P}_{t-1|t-1}\mathbf{F}_{t-1}^T + Q \tag{8}$$

- Update Step

$$\hat{\mathbf{y}}_t = \mathbf{y}_t - \mathbf{C}\hat{\mathbf{x}}_{t|t-1} \tag{9}$$

$$\mathbf{K}_t = \hat{\mathbf{P}}_{t|t-1}\mathbf{C}^T(\mathbf{C}\hat{\mathbf{P}}_{t|t-1}\mathbf{C}^T + R)^{-1} \tag{10}$$

$$\hat{\mathbf{x}}_{t|t} = \hat{\mathbf{x}}_{t|t-1} + \mathbf{K}_t\hat{\mathbf{y}}_t \tag{11}$$

$$\mathbf{P}_{t|t} = (\mathbf{I} - \mathbf{K}_t\mathbf{C})\hat{\mathbf{P}}_{t|t-1} \tag{12}$$

**Wavelet Filter**. A wavelet is a mathematical function used to divide a given function or continuous-time signal into different scale components. A wavelet transform is the representation of a function by wavelets. By shifting and scaling a mother wavelet function $\Psi(t)$, any function $f(t)$ can decomposed into coefficients with multi-resolution wavelets

$$\Psi_{a,b}(t) = \frac{1}{\sqrt{b}}\Psi(\frac{t-a}{b}), \gamma_{a,b} = \int f(t)\Psi_{a,b}(t)dt \tag{13}$$

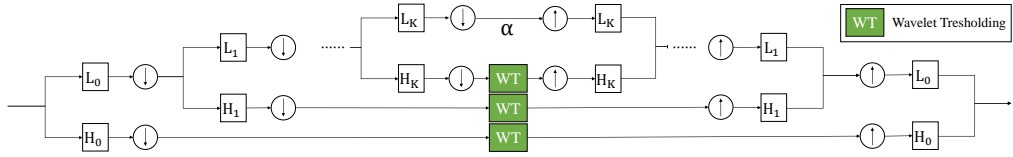

Figure 2: Pipeline for Wavelet Gradient Filtering for Gradient Decent. (1) Forward DWT to get a multi-resolution representation of the gradients (2) Thresholding and amplification on wavelet domain (3) Inverse DWT to the spatial domain.

where $a$ and $b$ are translation and scaling coefficients, $\gamma_{a,b}$ is the wavelet coefficients correspond to wavelet $\Psi_{a,b}(t)$. A discrete wavelet transform (DWT) is any wavelet transform for which the wavelets are discretely sampled. There are three basic steps to filtering signals using wavelets. (1) Decompose the signal using the DWT.

$$\Psi_{j,k}(t) = \frac{1}{\sqrt{2^j}}\Psi(\frac{t - k2^j}{2^j}), \gamma_{j,k} = \sum_t f(t)\Psi_{j,k}(t) \tag{14}$$

(2) Filter the signal in the wavelet space using thresholding operation $T$.

$$\gamma'_{j,k} = T(\gamma_{j,k}) \tag{15}$$

(3) Invert the filtered wavelet coefficients to reconstruct the filtered signal using the inverse DWT.

$$f'(t) = \sum_i \sum_k \gamma'_{j,k}\Psi_{j,k}(t) \tag{16}$$

The filtering of signals using wavelets is based on the idea that the DWT decomposes the signal into details and approximation parts. At some scales, the details contain mostly insignificant noise and can be removed or zeroed out using thresholding without affecting the signal.

### A.1.2 ENHANCING STOCHASTIC GRADIENT DESCENT

Stochastic gradient descent (SGD) serves as a means to minimize an objective function by iteratively adjusting parameters in opposition to the gradient direction. Despite its widespread use, vanilla SGD struggles to ensure optimal convergence of deep neural networks He et al. (2016); Dosovitskiy et al. (2020); Yang et al. (2022b); Hinton et al. (2015); Krizhevsky et al. (2017); Yang et al. (2022a) due to substantial gradient variance or the entrapment within suboptimal local minima. Momentum Qian (1999) addresses these issues by accelerating SGD with the addition of a $\gamma$ ratio from the previous update vector to the current one, simultaneously reducing gradient variance through a running average of past observations.

Further advancements include Adagrad Duchi et al. (2011), which assigns distinct learning rates to each parameter $\theta_i$ at every time step $t$, and its extensions Adadelta Zeiler (2012) and RM-Sprop Hinton et al., which aim to mitigate Adagrad's aggressive learning rate. Adaptive Moment Estimation (Adam) Kingma & Ba (2014) combines adaptive learning rates with a decaying average of past gradients, similar to momentum, and has been further expanded upon by variants such as AdaMax Kingma & Ba (2014) and Nadam Dozat (2016).

### A.1.3 REDUCING VARIANCE IN GRADIENT DESCENT

The estimation of noisy gradients using random data samples in SGD introduces significant variance, hindering optimal model convergence. Averaging techniques have emerged as a straightforward solution to this issue. Averaged Stochastic Gradient Descent (ASGD) Bottou (2012) exemplifies the simplest variance reduction approach by maintaining an average of a specific set of weight vectors. Stochastic Averaged Gradient (SAG) Schmidt et al. (2017) builds on this concept by updating the moving average of $n$ past gradients.

Other techniques, such as the variance reduction method by Wang et al. (2013), utilize control variates to enhance the noisy gradient, thereby reducing its variance. Additionally, Miller et al.

(2017) proposes a strategy for controlling the variance of the reparameterization gradient estimator in Monte Carlo variational inference (MCVI).

Drawing inspiration from signal processing, averaging functions as a basic low-pass filter, removing high-frequency components and noise from signals. Recently, the Kalman Optimizer Yang (2021) has demonstrated the application of the Kalman filter to gradient descent as a means of reducing gradient variance. Our Filtered Gradient Descent (FGD) builds upon these averaging techniques and the Kalman Optimizer, positing that if a simple low-pass filter is effective for gradient variance reduction, then more sophisticated filters may yield even better results.

## A.2 GRADIENT ESTIMATION WITH A FILTERING PROBLEM

Gradient estimation with a filter gets rid of the drawbacks of local gradient and data randomness by considering the historical states of the gradient, thus becoming more consistent and robust. We select four well-known filters (1) MA filter (2) AR filter (3) Kalman filter and (4) Wavelet filter as $F$ and would like to study how different filters affect the gradient estimation.

### A.2.1 SOLUTION 1: MOVING-AVERAGE FOR GRADIENT DECENT

When assuming that the observed and true gradients follow a moving-average model, we can approximate the true gradient $\hat{g}_{t|t}$ using an $MA(q)$ model

$$\hat{g}_{t|t} = g_t + \sum_{i=1}^{q} a_i g_{t-i}$$

$\{g_{t-i}\}_{i=1}^{q}$ is $q$ past stochastic gradients and $a_i$ is the weight for historical gradient $g_{t-i}$. We call this method Filter Gradient Decent with Moving-average filter (FGD-MA). In the following paper, we call the FGD-MA model with order $q$ as FGD-MA(q). It keeps a weighted average of $q$ past gradient observations.

### A.2.2 SOLUTION 2: AUTOREGRESSIVE FOR GRADIENT DECENT

When the autoregressive model best describes the relationship between the observed and true gradients, we may also estimate the true gradient $\hat{g}_{t|t}$ using an $AR(p)$ model

$$\hat{g}_{t|t} = g_t + \sum_{i=1}^{p} b_i \hat{g}_{t-i|t-i}$$

The $b_i$ is the weight corresponding to historical gradient estimation $\hat{g}_{t-i|t-i}$. We call this model Filter Gradient Decent with Autoregressive filter (FGD-AR). Interestingly, FGD-AR with order 1 turns out to be the momentum method that is commonly used in stochastic gradient descent

$$v_t = g_t + \gamma v_{t-1}$$

This shows that the momentum method is one special case for FGD-AR. FGD-AR extends the current momentum term to allow longer time dependency for gradient estimation. In the following paper, we call the FGD-AR model with order $p$ as FGD-AR(p). We also call the momentum method as FGD-AR(1).

### A.2.3 SOLUTION 3: KALMAN FILTER FOR GRADIENT DECENT

To reduce the variance of the gradient, we adopt a Kalman filter Yang (2021) to estimate the true state of the gradient upon a filtering problem. We assume the gradient follows a linear scholastic model

$$\hat{g}_t = \mathbf{F}_{t-1}\hat{g}_{t-1} + \mathbf{w}_{t-1}, \quad g_t = \mathbf{C}_{t-1}\hat{g}_t + \mathbf{v}_{t-1} \tag{17}$$

where $g_t$ is the observed gradient and $\hat{g}_t$ refers to real gradient. We adopt the Kalman filter to give the minimum mean-square estimate of real gradient $\hat{g}_{t|t}$ given previous state $\hat{g}_{t-1|t-1}$ and observation $g_t$ at time $t$.

$$\hat{g}_{t|t} = KF(g_t, \hat{g}_{t-1|t-1}) \tag{18}$$

By assuming that the gradient for each parameter is independent, we need to maintain a Kalman filter for each parameter. To reduce the computational burden, we simplified our method to a scalar Kalmen filter: $\mathbf{F}_t = \gamma I$, $\mathbf{C}_t = cI$, $Q = \sigma_Q^2 I$, $R = \sigma_R^2 I$, $\mathbf{P}_0 = p_0 I$, where $I$ is the identity matrix. This design cancels the time-consuming matrix multiplication, with only a scalar Kalman filter per parameter. We call this model Filter Gradient Descent with Kalman filter (FGD-K).

### A.2.4 SOLUTION 4: WAVELET GRADIENT DECENT

Wavelet Filtering is widely adopted in signal processing to remove certain frequency components of the signal. In order to better eliminate gradient noise, we consider the multi-resolution wavelet filter that soft-thresholds highpass coefficients at each level. We also intend to boost the low-frequency component of the gradient by amplifying the lowpass coefficient by a hyper-parameter of $\alpha$. Figure 2 explains the overall pipeline for our proposed Filter Gradient Descent with Wavelet filter (FGD-W). The procedure can be summarized into three steps

- *Forward DWT* We first apply K-level wavelet transformation for $L$ historical gradients to get multi-resolution representation of the gradient

$$(\boldsymbol{L}_K, \boldsymbol{H}_K, \ldots, \boldsymbol{H}_0) = DWT(\boldsymbol{g}_t, \ldots, \boldsymbol{g}_{t-L+1}) \tag{19}$$

$\boldsymbol{L}_i$ and $\boldsymbol{H}_i$ are the lowpass and highpass coefficient vectors at level $i$. Particularly, we assume independence for different elements of the gradient and maintain a 1D DWT for each parameter. In terms of implementation, we keep a $L$-length queue for each parameter to store $L$ historical gradient observations. Every time a new gradient $\boldsymbol{g}_t$ is computed, it is pushed in the queue while $\boldsymbol{g}_{t-L}$ is popped out.

- *Filter in wavelet domain* Soft-thresholding is performed on all highpass components $\boldsymbol{H}_i$ at each level with operation $T$ and amplify the lowpass band by a coefficient of $\alpha \geq 1$

$$\boldsymbol{H}_i' = T(\boldsymbol{H}_i), \quad \boldsymbol{L}_K' = \alpha \boldsymbol{L}_K \tag{20}$$

where $\boldsymbol{H}_i'$ and $\boldsymbol{L}_K'$ is the filtered highpass and low pass coefficients in the wavelets domain. For simplicity, we adopt the same threshold on all levels.

- *Inverse DWT* The filtered coefficients in wavelet domain are converted back to the spatial domain using inverse DWT

$$\boldsymbol{g}_{t|t}, \ldots, \boldsymbol{g}_{t-L+1|t} = IDWT(\boldsymbol{L}_K', \boldsymbol{H}_K', \ldots, \boldsymbol{H}_0')$$

## A.3 EXPERIMENT SETUP

### A.3.1 STUDY 1: NON-CONVEX OPTIMIZATION

For each experiment, we select a learning rate from $\eta = \{0.01, 0.03\}$ to minimize this function. The initial point is fixed to $(x, y) = (5, 5)$. For Adam, we use the default setting that $\beta_1 = 0.9, \beta_2 = 0.99$. For FGD-MA and FGD-AR, we select $a_1 = b_1 = 0.9$ for order 1 model and $a_1 = b_1 = 0.8, a_2 = b_2 = 0.1$ for order 2. For FGD-K, we select $\gamma = 2, c = 1, \sigma_Q^2 = 0.01, \sigma_R = 2$. For FGD-W, we use Daubechies 4 (db4) wavelet and $K = 3$ and $L = 8$. We apply soft-thresholding on each highpass band with a threshold of 0.2 and select $\alpha = 5$. Each Algorithm runs for $T = 500$ iterations.

### A.3.2 STUDY 2: MULTI-LAYER PERCEPTRON

We establish a 3-layer neural network with 784 input units, 10 hidden units, and 10 output units. All training images are augmented with zero-padding of 2 pixels and random cropped to $28 \times 28$. Each image is flattened as a 1-dimensional vector. We set the initial learning rate $\eta = \{0.1, 0.01, 0.001\}$ and mini-batch size $batchsize = \{4, 16, 64\}$. The learning rate is multiplied by 0.7 every epoch. For Adam, we use the default setting that $\beta_1 = 0.9, \beta_2 = 0.99$. For FGD-MA and FGD-AR, we select $a_1 = b_1 = 0.9$ for order 1 model and $a_1 = b_1 = 0.8, a_2 = b_2 = 0.1$ for order 2. For FGS-K, we select $\gamma = 5, c = 1, \sigma_Q^2 = 0.01, \sigma_R = 2$. For FGD-W, we use Daubechies 4 (db4) wavelet and $K = 3$ and $L = 16$. We apply soft-thresholding on each highpass band with a threshold of 0.2 and select $\alpha = 5$. We train the network for 10 epochs. The network is trained with a cross-entropy loss.

