# OpenReview forum: "Towards Stochastic Gradient Variance Reduction by Solving a Filtering Problem"
_ICLR.cc/2023/TinyPapers — Submitted to Tiny Papers @ ICLR 2023_

### Official Review · Reviewer_WndZ · 2023-03-23

**Confidence:** 3

**Summary Of Contributions:**

The paper propose e Filter Gradient Decent (FGD), a family of stochastic optimization algorithms that makes the consistent estimation of the local gradient by solving an adaptive filtering problem. The paper shows that FDG can reduce the variance of the SGD and improve training a classifier on MNIST compared to traditional momentum-based methods.

**Rating:**

High Potential (HP): a submission which meets the reviewing criteria and has potential to make an impact on the field

**Strengths And Weaknesses:**

# Strengths
- Various types of filters are proposed including Moving Average filter (FGD-MA), Auto-regressive filter (FGD-AR), Kalman filter (FGD-K), and Wavelet filter (FGD-W).
- Experiments show that FGD significantly improves the classification accuracy of MNIST compared to the conventional SGD and Adam optimizer.

# Weaknesses
- Experiments on more large-scale datasets should be considered e.g., CIFAR10 and CIFAR100.

**Suggested Changes:**

Experiments on more large-scale datasets should be considered e.g., CIFAR10 and CIFAR100.

---

### Comment · Area_Chair_mcyN · 2023-05-30
**Invite to archive**

Hi all,

This paper was initially invited to archive.  The author(s) appear to have updated the appendix considerably to address some reviewer feedback.  Therefore, I still recommend that this paper is archived.

Thanks,
AC mcyN

---

### Comment · Area_Chair_mcyN · 2023-06-02
**Update for archival**

The current manuscript is missing the URM statement.  Please re-upload a version of the paper with the URM statement completed (this can be on the third page).

```Please include this URM Statement section at the end of the paper but before the references before. In your anonymized submission, we recommend stating ``The authors acknowledge that at least one key author of this work meets the URM criteria of ICLR 2023 Tiny Papers Track.'' For the camera ready version, we ask authors to identify which author(s) meet the URM criteria, e.g., ``Author TFB meets the URM criteria of ICLR 2023 Tiny Papers Track.'' The authors are also welcome to come up with their own phrases to affirm meeting this criteria.```

---

### Comment · Area_Chair_mcyN · 2023-06-06
**Ready for archival**

This work meets the threshold for archival, contents the URM statement and is deanonymized.

---

### Meta-Review · Area_Chair_mcyN · 2023-04-06

**Recommendation:** Invite to archive
**Confidence:** 4

**Metareview:**

This paper introduces the concept of gradient estimation as a filtering problem.  The paper itself is fairly well written, and on the surface, the connection is novel.

However, I am currently unsure if the discussion, analysis and experimental results presented are ``complete''.  I believe moving averages of gradients are implemented via momentum in most commonly used optimisers.  While links to Kalman filtering, AR processes etc are nice, I query if they are subsumed into existing gradient methods?  For instance, in Table 1 I would look to see the other parameters of ADAM swept over as well (such as the momentum) to see if there are links there.  This makes me wonder if the new estimators in Table 1 have been tuned more aggressively than the baseline.

I have similar queries for the synthetic example.  ADAM appears to fall in to a local minima and then terminate, suggesting that the hyperparameters are not well-tuned.  In some sense as well, ADAM is ``doing the right thing'' from the initial value, whereas other methods are benefiting from noise early in the estimation procedure to jump out of the basin of attraction.

I also wonder how the Kalman filter is applied in high dimensions, as it requires computing an inverse, which is computationally intractable for large networks (and slower than SGD).  I see you make some diagonal assumptions in the supplement, but I would still like to see some quantitive timings and discussions of the limitations of a diagonal assumption.  I also query how fast the wavelet method is, and how difficult to tune to wavelet method is.

**Summary:**

This work tries to draw links between filtering and optimisation.  The link is appealing, but I do not believe the method is sufficiently fleshed out to quite ready for inclusion yet.

**Comments And Feedback To The Authors:**

I have several comments for the authors:
- A more thorough evaluation over the hyperparameters of the ADAM baseline, especially given that you are comparing to the momentum aspect of this.
- Different optimisers would also be good to compare to -- Eg ADAgrad, RMSprop etc.  ADAM does have some known pathologies on simple problems.
- More qualitative discussion of how your method relates to momentum-based optimisers.
- Maybe pick just one or two variants of FGD?  There are a lot of columns and lines on the results and it makes it difficult to tease apart what is a significant result.
- For the toy example, please provide quantitive results across random initialisations from the range $[-6, 6]^2$.  This will help combat the criticism that this one initialisation is just particularly bad for ADAM.
- It is generally best to take a neutral tone in written work, eg. avoiding descriptions such as ``bold''.
- I would also like to see some experiments studying the bias and variance of the estimator.  This can be done by computing a distance between the distribution over the error in the gradient estimate across the parameter domain, or by plotting a histogram of the errors across the domain.  From the experiments in Figure 1, it looks like your methods are actually *higher* variance, and maybe even biased, compared to ADAM.


There are also several typographical mistakes, I will outline some here.
- The first word should be capitalized.
- Only proper nouns should be capitalised (Filtering Gradient Descent -> filtering gradient descent, Experiment for Numerical Non-convex optimisation -> Experiment for numerical non-convex optimisation).
- Figures should be floated to the top of pages as opposed to inlined in text.
- Text on figures should be about the same size as the body text.
- Bold-faced text is unnessacary.

I hope this feedback helps.  I think there is a valuable contribution/idea in here somewhere, it just isn't quite ready yet.


**Reason For Not Giving A Higher Recommendation:**

The idea is interesting, but I do not believe it is currently quite ready for inclusion, citing insufficient evaluation over the core aspect of the baseline.  With further evaluation, this work could be very impactful.

**Reason For Not Giving A Lower Recommendation:**

N/A

---

### Decision · Program_Chairs · 2023-04-08

Invite to archive